# CCIL: Continuity-based Data Augmentation for Corrective Imitation Learning

**Liyiming Ke**[*], **Yunchu Zhang**[*], **Abhay Deshpande, Siddhartha Srinivasa, Abhishek Gupta**
Paul G. Allen School of Computer Science and Engineering, University of Washington
{kayke,yunchuz,abhayd,siddh,abhgupta}@cs.washington.edu

**Abstract:**

We present a new technique to enhance the robustness of imitation learning methods by generating corrective data to account for compounding error and disturbances. While existing methods rely on interactive expert labeling, additional offline datasets, or domain-specific invariances, our approach requires minimal additional assumptions beyond access to expert data. The key insight is to leverage local continuity in the environment dynamics to generate corrective labels. Our method first constructs a dynamics model from the expert demonstration, enforcing local Lipschitz continuity in the learned model. In locally continuous regions, this model allows us to generate corrective labels within the neighborhood of the demonstrations but beyond the actual set of states and actions in the dataset. Training on this augmented data enhances the agent's ability to recover from perturbations and deal with compounding error. We demonstrate the effectiveness of our generated labels through experiments over a variety of robotics domains.

## 1 Introduction

Deploying imitation learning for real-world robotics requires a vast amount of data. With sufficient data coverage, the simple and practical behavior cloning method has shown tremendous success [1, 2, 3]. However, when robotic policies encounter states not covered in the expert dataset due to sensor noise, stochastic environments, covariate shift [4, 5], they can act unpredictably and dangerously. For widespread deployment of robotic applications, we need a solution that ensures the robustness of imitation learning policies even when encountering unfamiliar states. Many approaches rely on augmenting the dataset, either through interactive experts [4, 6] or understanding system invariances [2, 7, 8]. However, these techniques can be costly or infeasible [9, 10], leaving behavior cloning, which only requires expert demonstrations, as the prevalent choice [3, 9, 11, 12, 13].

For applicability, we propose an augmentation method for robust imitation learning based *solely on expert demonstrations*. We identify a crucial feature of dynamic systems that is under-exploited: the inherent continuity in dynamic systems. Even though system dynamics may have complex transitions and representations, they need to adhere to the laws of physics and exhibit some level of continuity. While realistic dynamical systems can contain discontinuity in certain portions of the state space, the subset of state space or latent space that exhibits local continuity can be a powerful asset.

Armed with this structure, we aim to develop corrective labels to redirect agents from unfamiliar states back to familiar ones. The structure of local Lipschitz continuity allows an appropriately regularized dynamics model to be effective even outside the expert data zone. We propose a practical algorithm, **CCIL**, leveraging local **C**ontinuity in dynamics to generate **C**orrective labels for **I**mitation **L**earning. It first trains an appropriately-regularized dynamics function and then use it to synthesize *corrective* labels, to mitigate compounding errors in imitation learning. In summary, our contributions are:

7th Conference on Robot Learning (CoRL 2023), Atlanta, USA.

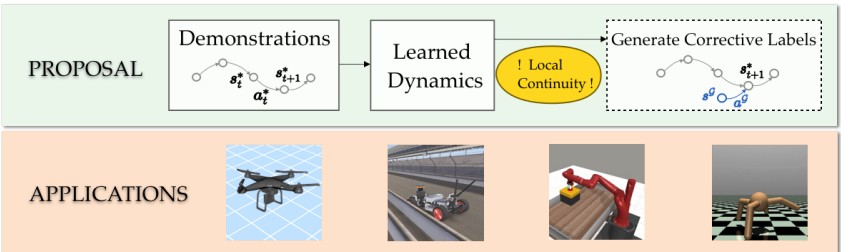

Figure 1: **Overview of CCIL.** To enhance robustness of imitation learning, we propose to augment the dataset with synthetic corrective labels. We leverage the local continuity in the dynamics, learn a regularized dynamics function and generate corrective labels *near* the expert data support. We provide theoretical guarantees on the quality of the generated labels. We present empirical evaluations CCIL over 4 distinct robotic domains to showcase CCIL capability to improve imitation learning agents' robustness to disturbances.

- **Problem Formulation:** A formalized concept of corrective labels to enhance robustness for imitation learning. (Sec. 2.1).
- **Practical Algorithm:** The **CCIL** method, leveraging expert demonstrations and dynamics continuity to generate corrective labels.
- **Theoretical Guarantees:** Exploration of local continuity's role in expanding a model's reach beyond the expert dataset. Present practical methods to enforce desired local smoothness while fitting a dynamics function while accommodating discontinuity (Sec. 2.2). Provide a theoretical bound on the quality of the model in this area and the generated labels (Sec. 2.3).
- **Extensive Empirical Validation:** Tests over 4 distinct robotic domains across 14 tasks ranging from classic control, drone navigation, high-dimensional driving, manipulation and locomotion to showcase our proposal's ability to enhance the robustness of imitation learning agents (Sec. 3).

## 2 Generative Corrective Labels via Continuity in Dynamics

We first define the desired *high quality corrective* labels to make IL more robust in Sec. 2.1. To generate the desired labels using learned dynamics function, the dynamics need to exhibit local continuity and we present a method to train a locally Lipschitz-bounded dynamics model in Sec. 2.2. In Sec. 2.3, we illustrate how to use a learned model to obtain corrective labels. We instantiate these insights into a practical algorithm - **CCIL**, more details in Appendix. D.

### 2.1 Corrective Labels Formulation

Our goal is to enable IL to be robust to new states it might encounter by generating a set of state-action-state triplets $(s^{\mathcal{G}}, a^{\mathcal{G}}, s_{\text{next}})$ that are *corrective*. Intuitively, if executing action $a^{\mathcal{G}}$ in state $s^{\mathcal{G}}$ on the system dynamics $f$ can bring the agent back to a state $s_{\text{next}}$ that is in support of the expert data distribution, then $(s^{\mathcal{G}}, a^{\mathcal{G}})$ is a corrective state-action pair. By bringing the agent back to the "known" expert data distribution, where the policy is likely to be successful, the labels provide corrections to disturbances. However, the true system dynamics, $f$, is unknown. Instead, we have only an approximation of the dynamics function, $\hat{f}$. We define corrective labels using an approximate model:

**Definition 2.1. [High Quality Corrective Labels under Approximate Dynamic Models].** $(s^{\mathcal{G}}, a^{\mathcal{G}}, s_{\text{next}})$ is a corrective label if $\|[s^{\mathcal{G}} + \hat{f}(s^{\mathcal{G}}, a^{\mathcal{G}})] - s_{\text{next}}\| \le \epsilon_c$, w.r.t. an approximate dynamics $\hat{f}$. Such a label is "high-quality" if the approximate dynamics function also has bounded error w.r.t. the ground truth dynamics function at the given state action $\|f(s^{\mathcal{G}}, a^{\mathcal{G}}) - \hat{f}(s^{\mathcal{G}}, a^{\mathcal{G}})\| \le \epsilon_{co}$.

Intuitively, the high-quality corrective labels represent the learned dynamics model's best guess at bringing the agent back into the support of expert data. However, an approximate dynamics model is only trustworthy in a certain region of the state space, i.e., where the predictions of $\hat{f}$ approximately match the true dynamics of the system.

**Algorithm 1** CCIL: **C**ontinuity-based data augmentation for **C**orrective labels for **I**mitation **L**earning

1: **Input:** $\mathcal{D}* = (s_i^*, a_i^*, s_{i+1}^*)$
2: **Initialize:** $D^{\mathcal{G}} \leftarrow \varnothing$
3: Learn a Dynamics Function $\hat{f}$
4: **for** $i = 1..n$ **do**
5:    $(s_i^{\mathcal{G}}, a_i^{\mathcal{G}}) \leftarrow$ GenLabels $(s_i^*, a_i^*, s_{i+1}^*)$
6:    **if** $||s_i^{\mathcal{G}} - s_i^*|| < \epsilon$ **then**
7:      $\mathcal{D}^{\mathcal{G}} \leftarrow \mathcal{D}^{\mathcal{G}} \cup (s_i^{\mathcal{G}}, a_i^{\mathcal{G}})$
8:    **end if**
9: **end for**

10: **Function** Learn a Dynamics Model $\hat{f}$
11:    Optimize a chosen objective from Sec. 2.2
12: **Function** Gen DisturbedAction Labels
13:    $a_i^{\mathcal{G}} \leftarrow a_i^* + \Delta, \Delta \sim \mathcal{N}(0, \Sigma)$
14:    $s_i^{\mathcal{G}} \leftarrow \arg\min_{s_i^{\mathcal{G}}} ||s_i^{\mathcal{G}} + \hat{f}(s_i^{\mathcal{G}}, a^{\mathcal{G}}) - s_{i+1}^*||$
15: **Function** Gen BackTrack Labels
16:    $a_i^{\mathcal{G}} \leftarrow a_i^*$
17:    $s_i^{\mathcal{G}} \leftarrow \arg\min_{s_i^{\mathcal{G}}} ||s_i^{\mathcal{G}} + \hat{f}(s_i^{\mathcal{G}}, a_i^{\mathcal{G}}) - s_i^*||$

## 2.2 Learning Locally Continuous Dynamics Functions from Data via Slack Variable

Our core insight is to leverage the inherent continuity in dynamic systems. When the dynamics are locally Lipschitz bounded, small changes in state and actions yield minimal changes in the resulting transitions. A trained dynamics function that is correspondingly locally smooth would allow us to extrapolate to the states and actions that are in close proximity to the expert demonstration - a region that we can trust the learned model.

We follow the framework of model-based reinforcement learning to train our dynamics model using expert data [14]. Critically, we ensure the learned dynamics model would contain local continuity. In face of *discontinuity*, we present a practical approach to fit a dynamics model that (1) enforces as much local continuity as permitted by the data and (2) discards the discontinuous regions when generating labels. Inspired by SVM [15], we can explicitly allow for a small amount of discontinuity in the learned dynamics model in the same way that slack variables are modeled in optimization problems. We can formulate the dynamics model's learning objective in App. D.1.

## 2.3 Generating High Quality Corrective Labels Using Locally Continuous Dynamics Models

By leveraging local continuity, we can use a trained model to confidently navigate unfamiliar states within a range that exceeds the coverage of expert data. When the local dynamics function is bounded by a Lipschitz constant w.r.t. the state-action space, we query the dynamics function with a perturbed state or action from the expert demonstration (e.g., adding noise). The prediction from the dynamics model can be trusted when the perturbation is small. A perturbed state or action that, according to the learned model, returns the agent to expert support is a strong candidate for corrective labels. Specifically, we will perturb the state-action in the neighborhood of $s^*$ and $a^*$ to generate corrective labels. We thus outline two techniques to generate corrective labels.

**Technique 1: Backtrack Label.** For every expert label $s_t^*, a_t^*$, we propose to find a different state $s_{t-1}^{\mathcal{G}}$ that can arrive at $s_t^*$. To do so, we reformulate the optimization problem to use the expert action, $a_t^*$, and then leverage the Lipschitz continuity on states to find the particular $s_{t-1}^{\mathcal{G}}$:

$$s_t^* - \hat{f}(s_{t-1}^{\mathcal{G}}, a_t^*) - s_{t-1}^{\mathcal{G}} = 0. \tag{1}$$

**Technique 2: Disturbed Action.** Given demonstration $(s_t^*, a_t^*, s_{t+1}^*)$, we ask: Is there an action $a^{\mathcal{G}}$ that slightly differs from the demonstrated action $a_t^*$, i.e., $a^{\mathcal{G}} = a_t^* + \Delta$, that can bring an unknown state $s^{\mathcal{G}}$ to the same expert state $s_{t+1}^*$? Formally, we sample a small action noise $\Delta$ and solve for $s_t^{\mathcal{G}}$:

$$s_t^{\mathcal{G}} + \hat{f}(s_t^{\mathcal{G}}, a_t^* + \Delta) - s_{t+1}^* = 0. \tag{2}$$

Both techniques allow us to generate corrective labels with *bounded error*: executing the generated labels in the ground truth dynamics has reasonable chance of bringing the agent back to expert states. We present the bounds and proofs in App. C.1 and C.2. To solve the root-finding equations (Eq. 1 and 2), we turn them into optimization problems (App. D.4). We also use rejection sample to filter out generated labels that are prone to error App. D.5. We now instantiate a practical version of our proposal, **CCIL** (using **C**ontinuous dynamics to generate **C**orrective labels for **I**mitation **L**earning), as shown in Alg 1.

## 3   Experiments

We evaluate CCIL over four distinct robotic domains (Fig. 1) over 14 tasks. Our evaluation spans over classic control, drone navigation that is sensitive to noise disturbance, car racing that employes high-dimensional Lidar signals as state input, MuJoCo locomotion and MetaWorld manipulation which contain various forms of contacts and discontinuity. We summarize the highlights from our experiments and defer to App. E for elaboration and details to reproduce the experiments.

**Q1**: We validate the theoretical contributions on the classic Pendulum problem. We observe the empirical errors of the generated labels abide by our theoretically derived bound.
**Q2**, we compare CCIL with classic behavior cloning (BC) and NoisyBC [9]. We evaluate the trained agent's robustness by injecting small observation and action disturbances. We observe that, on the Pendulum, Drone, and Car tasks, CCIL greatly improves BC. On locomotion and manipulation tasks, CCIL exhibits advantages on 4 of 8 tasks, only losing to NoiseBC on 1 of the 8 tasks.
**Q3**, we examine how the performance of CCIL would fluctuate facing various forms of discontinuity. We compare CCIL on Pendulum versus Discontinuous Pendulum (by placing a wall). We observe that for both tasks, CCIL can improve BC's performance, but the boost was smaller for discontinuous Pendulum. Noticeably, CCIL could improve BC performance for the racing task with high-dimensional Lidar-inputs that contains complex forms of discontinuity. Finally, we find that CCIL was able to achieve at least comparable performance to behavior cloning for tasks with lots of contacts that make training of dynamics model challenging (e.g., MetaWorld manipulation).

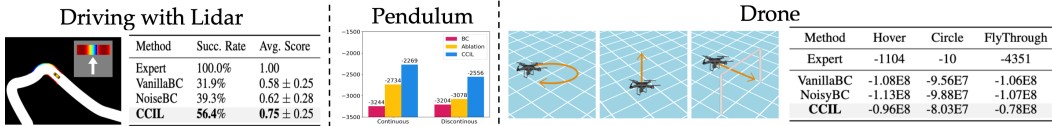

Figure 2: Evaluation on the Driving, Discontinuous Pendulum and Drone Tasks.

Table 1: Evaluation results for Mujoco and Metaworld tasks with noise disturbances. We list the expert scores in a noise-free setting for reference. In the face of varying discontinuity from contacts, CCIL remains the leading agent on 4 out of 8 tasks (Hopper, Walker, HalfCheetah, CoffeePull). Comparing CCIL with BC: across all tasks, CCIL can outperform vanilla behavior cloning or at least achieve comparable performance.

| | **Mujoco** | | | | **Metaworld** | | | |
|---|---|---|---|---|---|---|---|---|
| | Hopper | Walker | Ant | Halfcheetah | CoffeePull | ButtonPress | CoffeePush | DrawerClose |
| Expert | 3234.30 | 4592.30 | 3879.70 | 12135.00 | 4409.95 | 3895.82 | 4488.29 | 4329.34 |
| VanillaBC | 1983.98 ± 672.66 | 1922.55 ± 1410.09 | 2965.20 ± 202.71 | 1798.98 ± 791.89 | 3552.59 ±233.41 | **3693.02** ± 104.99 | 1288.19± 746.37 | 3247.06 ± 468.73 |
| NoiseBC | 1563.56 ± 1012.02 | 2893.21 ± 1076.89 | **3776.65** ± 442.13 | 2044.24 ±291.59 | 3072.86 ± 785.91 | **3663.44**±63.10 | 2551.11± 857.79 | 4226.71± 18.90 |
| CCIL | **2631.25** ± 303.86 | **3538.48** ± 573.23 | 3338.35 ± 474.17 | **8893.81** ± 472.70 | **4168.46** ± 192.98 | **3775.22**±91.24 | **2484.19**± 976.03 | 4145.45± 76.23 |

## 4   Conclusion

We propose CCIL to generate corrective labels for imitation learning by leveraging local continuity in environmental dynamics. Our method uncovers two new ways to produce high-quality labels for out-of-distribution states and robust imitation learning. While the assumption is that systems exhibit some local continuity, we've confirmed the effectiveness of our method on various simulated robotics environments, including drones, driving, locomotion and manipulation. Empirically, CCIL achieves at least comparable results with the baseline IL methods and exhibits clear advantage when the environmental dynamics is easier to learn.

We believe our approach opens the door for many exciting avenues of future research. To quantify and capture the local continuity for robot systems' dynamics can help make imitation learning more robust. Extending our proposal to real robot and to high-dimensional state (e.g., pixel-based image) could lead to exciting algorithms to alleviate the data hunger in imitation learning. Examining how best to fit dynamics models could not only benefit our proposal, but also provide insights to help other model-based learning algorithms (e.g., model-based reinforcement learning).

## 5 Acknowledgments

We thank Jack Umenberger for enlightening discussions; Khimya Khetarpal, Sidharth Talia and folks at the WEIRDLab at UW for feedback on the paper. This work was (partially) funded by the National Science Foundation NRI (#2132848) and CHS (#2007011), the Office of Naval Research (#N00014-17-1-2617-P00004 and #2022-016-01 UW), and Amazon.

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

# A Related Work

**Imitation Learning (IL) and Data Augmentation.** With access only to expert demonstrations, behavior cloning remains a strong empirical baseline for imitation learning [1]. This method formulates IL as a supervised learning problem and has a plethora of data augmentation methods. However, previous augmentation methods mostly leverage expert or some form of invariance [16, 2, 7, 8], which is a domain-specific property. Ke et al. [9] explored injecting noise to the expert demonstrations, similar to our approach, but lacked theoretical insights and general guidelines for choosing noise parameters. In contrast, our proposal leverages the local continuity in the dynamics function, agnostic to domain-knowledge, and provides theoretical guarantees on the quality of augmented data.

**Mitigating Covariate Shift in Imitation Learning.** Compounding errors push the agent astray from expert demonstrations. Prior works addressing the covariate shift often request additional information. Methods like DAGGER [4], LOLS [17], DART [6] and AggrevateD [18] use interactive experts, while GAIL [19], SQIL [20] and AIRL [21] sample more transitions in the environment to minimize the divergence of states distribution between the learner and expert [22, 23]. Offline Reinforcement Learning methods like IQL [24], MOREL [25] and CQL [26] demand a ground truth reward function. MILO [27] also learns a dynamics function to mitigate covariate shift in imitation learning but requires a large batch of sub-optimal offline data to learn a high-fidelity dynamics function. In contrast, our proposal is designed for learning from demonstration paradigms without requiring additional data or feedback, complementing existing IL methods.

**Locally Lipschitz Continuity in Dynamics.** Classical control methods often assume local Lipschitz continuity in the dynamics to guarantee the existence and uniqueness of solutions to differential equations. For example, the widely adopted $\mathcal{C}^2$ assumption in optimal control theory [28] and the popular control framework iLQR [29]. This assumption is particularly useful in the context of nonlinear systems and are prevalent in modern robot applications [30, 31, 32]. However, these methods require a pre-specified dynamics model, while this work focuses on learning a locally continuous dynamics model from data.

**Learning Dynamics using Neural Networks.** Fitting a dynamics function from data is an active area of research [33, 34, 35]. Ensuring local continuity in the trained dynamics can be challenging. Previous works enforced Lipschitz continuity in training neural networks [36, 37, 38] but not for dynamics functions with physical implications. Shi et al. [39] learned smooth dynamics functions via enforcing *global* Lipschitz bounds and is only demonstrated on the problem of drone landing. Pfrommer et al. [40] learned a smooth model to accommodate fictional contacts for manipulation. We provide a novel method to effectively learn locally smooth dynamics functions for generic environments. Further, Khetarpal et al. [41], Zhang et al. [42], Zhu et al. [43] are actively researching on learning compact world models from high-dimensional inputs or to capture invariance. Progress in this direction could enable our proposal to leverage the continuity in the learned latent space and extend to complex representation of states.

# B Generating Corrective Labels with Known Dynamics Function

With a known dynamics function, we show an example algorithm that one can use to generate $N$ corrective labels (as defined in Sec 2.1). The algorithm first trains a behavior cloning policy, samples test time roll out trajectories from the learned policy, and then derives labels using a root finding solver.

---

**Algorithm 2** Generating Corrective Labels using Dynamics Function

---

1: **Input** Expert Data $\mathcal{D}* = (s_i^*, a_i^*, s_{i+1}^*)$.
2: **Input** Dynamics function $f(s^{i+1} | s^i, a^i)$.
3: **Input** Parameter $N$.
4: Initialize $\mathcal{D}^{\mathcal{G}} \leftarrow \varnothing, \mathcal{S}' \leftarrow \varnothing$
5: $\hat{\pi} = \arg\min_{\hat{\pi}} -\mathbb{E}_{s_i^*, a_i^*, s_{i+1}^* \sim \mathcal{D}*} \log(\hat{\pi}(a_i^* \mid s_i^*))$
6: **for** $i \in 1..N$ **do**
7: $\quad s_0^{\mathcal{G}} \sim P_0, s_{j+1}^{\mathcal{G}} \sim f(s_j^{\mathcal{G}}, \pi(s_j^{\mathcal{G}})), \mathcal{S}' \leftarrow \mathcal{S}' \cup \{s_j^{\mathcal{G}}\}$
8: **end for**
9: **for** $i \in 1..N$ **do**
10: $\quad a_i^{\mathcal{G}} \leftarrow \arg\min_{a^{\mathcal{G}}} ||[s_i^{\mathcal{G}} + f(s_i^{\mathcal{G}}, a_i^{\mathcal{G}})] - s_k^*||$
11: $\quad \mathcal{D}^{\mathcal{G}} \leftarrow \mathcal{D}^{\mathcal{G}} \cup (s_i^{\mathcal{G}}, a_i^{\mathcal{G}})$
12: **end for**
13: **return** $\mathcal{D}^{\mathcal{G}}$

---

# C Proofs

## C.1 Quality of backtrack labels

**Theorem C.1.** *When the dynamics model has a bounded training error $\epsilon$ on the training data, under the assumption that the dynamics functions $f$ and $f'$ are locally Lipschitz continuous w.r.t. state, then*

$$\left\| f\left(s_t^{\mathcal{G}}, a_{t+1}^*\right) - \hat{f}\left(s_t^{\mathcal{G}}, a_{t+1}^*\right) \right\| \leq \epsilon + (K_1 + K_2) \left\| s_t^{\mathcal{G}} - s_{t+1}^* \right\|. \tag{3}$$

**Notation**. Let $f$ be the ground truth 1-step residual dynamics model, and let $\hat{f}$ be the learned approximation of $f$.

**Assumptions**:

1. The estimation error of the learned dynamics model *at the training data* is bounded.
   $\left\| f(s_{t+1}^*, a_{t+1}^*) - \hat{f}(s_{t+1}^*, a_{t+1}^*) \right\| \leq \epsilon$.

2. $\hat{f}$ is locally $K_1$-Lipschitz in state around data points:
   $\left\| \hat{f}(s_t^{\mathcal{G}}, a_{t+1}^*) - \hat{f}(s_{t+1}^*, a_{t+1}^*) \right\| \leq K_1 \left\| s_t^{\mathcal{G}} - s_{t+1}^* \right\|$.

3. $f$ is locally $K_2$-Lipschitz in state around data points:
   $\left\| f(s_t^{\mathcal{G}}, a_{t+1}^*) - f(s_{t+1}^*, a_{t+1}^*) \right\| \leq K_2 \left\| s_t^{\mathcal{G}} - s_{t+1}^* \right\|$.

**Proof**:

$$\left\| f(s_t^{\mathcal{G}}, a_{t+1}^*) - \hat{f}(s_t^{\mathcal{G}}, a_{t+1}^*) \right\|$$

$$= \left\| f(s_t^{\mathcal{G}}, a_{t+1}^*) - f(s_{t+1}^*, a_{t+1}^*) + f(s_{t+1}^*, a_{t+1}^*) - \hat{f}(s_{t+1}^*, a_{t+1}^*) + \hat{f}(s_{t+1}^*, a_{t+1}^*) - \hat{f}(s_t^{\mathcal{G}}, a_{t+1}^*) \right\|$$

$$\leq \left\| f(s_t^{\mathcal{G}}, a_{t+1}^*) - f(s_{t+1}^*, a_{t+1}^*) \right\| + \left\| f(s_{t+1}^*, a_{t+1}^*) - \hat{f}(s_{t+1}^*, a_{t+1}^*) \right\| + \left\| \hat{f}(s_{t+1}^*, a_{t+1}^*) - \hat{f}(s_t^{\mathcal{G}}, a_{t+1}^*) \right\|$$

$$\leq \epsilon + (K_1 + K_2) \left\| s_t^{\mathcal{G}} - s_{t+1}^* \right\|$$

**Remark** Our assumption about the error of the learned dynamics model is not a global constraint but simply requires the model to have prediction small error *on the training data*. Our proof leverages simple triangle inequality and is valid only near the expert data support.

## C.2 Quality of Noisy Labels

**Theorem C.2.** *Given $s_{t+1}^* - \hat{f}(s_t^{\mathcal{G}}, a_t^* + \Delta) - s_t^{\mathcal{G}} = \epsilon$ and that the dynamics function $f$ is locally Lipschitz continuous w.r.t. actions and states with Lipschitz constants $K_1$ and $K_2$, respectively, then*

$$||f(s_t^{\mathcal{G}}, a_t^* + \Delta) - \hat{f}(s_t^{\mathcal{G}}, a_t^* + \Delta)|| \leq K_1 ||\Delta|| + (1 + K_2)\epsilon. \tag{4}$$

**Notation**. Let $f$ be the ground truth 1-step residual dynamics model, and let $\hat{f}$ be the learned approximation of $f$.

**Assumptions**

1. $f$ is locally $K_1$-Lipschitz in *action*: $\left\| f(s_t^{\mathcal{G}}, a_t^*) - f(s_t^{\mathcal{G}}, a_t^* + \Delta) \right\| \leq K_1 \left\| \Delta \right\|$.

2. $f$ is locally $K_2$-Lipschitz in *state*: $\left\| f(s_t^{\mathcal{G}}, a_t^*) - f(s_t^*, a_t^*) \right\| \leq K_2 \left\| s_t^{\mathcal{G}} - s_t^* \right\|$.

3. Using rejection sampling, we can enforce $\left\| s_t^{\mathcal{G}} - s_t^* \right\| \leq \epsilon_{rej}$.

4. Given that the generated labels come from a root solver:

$$s_{t+1}^* - \hat{f}(s_t^{\mathcal{G}}, a_t^* + \Delta) - s_t^{\mathcal{G}} = \epsilon_{opt} \text{ where } \epsilon_{opt} \to 0$$
$$s_t^* + f(s_t^*, a_t^*) - \hat{f}(s_t^{\mathcal{G}}, a_t^* + \Delta) - s_t^{\mathcal{G}} = \epsilon_{opt}$$
$$f(s_t^*, a_t^*) - \hat{f}(s_t^{\mathcal{G}}, a_t^* + \Delta) = s_t^{\mathcal{G}} - s_t^* + \epsilon_{opt}$$

**Proof**

$$\left\| f(s_t^{\mathcal{G}}, a_t^* + \Delta) - \hat{f}(s_t, a_t^* + \Delta) \right\|$$
$$= \left\| f(s_t^{\mathcal{G}}, a_t^* + \Delta) - f(s_t^{\mathcal{G}}, a_t^*) + f(s_t^{\mathcal{G}}, a_t^*) - f(s_t^*, a_t^*) + f(s_t^*, a_t^*) - \hat{f}(s_t, a_t^* + \Delta) \right\|$$
$$\leq \left\| f(s_t^{\mathcal{G}}, a_t^* + \Delta) - f(s_t^{\mathcal{G}}, a_t^*) \right\| + \left\| f(s_t^{\mathcal{G}}, a_t^*) - f(s_t^*, a_t^*) \right\| + \left\| f(s_t^*, a_t^*) - \hat{f}(s_t, a_t^* + \Delta) \right\|$$
$$\leq K_1 \left\| \Delta \right\| + K_2 \left\| s_t^{\mathcal{G}} - s_t^* \right\| + \left\| s_t^{\mathcal{G}} - s_t^* \right\| + \left\| \epsilon_{opt} \right\|$$
$$\leq K_1 \left\| \Delta \right\| + (1 + K_2) \cdot \epsilon_{rej} + \left\| \epsilon_{opt} \right\|$$

When the root solver yields a solution with $\left\| \epsilon_{opt} \right\| = 0$, we have $\leq K_1 \left\| \Delta \right\| + (1 + K_2) \cdot \epsilon_{rej}$

# D   Details for CCIL

Our proposed framework for generating corrective labels, **CCIL**, takes three steps:

1. **Learn a dynamics model**: fit a dynamics model $\hat{f}$ that is locally Lipschitz continuous.
2. **Generate labels**: solve a root-finding equation in Sec. 2.3 to generate labels.
3. **Augment the dataset and train a policy**: We use behavior cloning for simplicity to train a policy.

## D.1   Learning a locally Lipschitz continuous dynamics model

There are many function approximators to learn a model. For example, using Gaussian process can produce smooth dynamics model but might have limited scalability when dealing with large amount of data. In this paper we demonstrate examples of using a neural network to learn the dynamics model. In practice, we write down the dynamics learning loss:

$$\arg \min_{\hat{f}} \mathbb{E}_{s_j^*, a_j^*, s_{j+1}^* \sim \mathcal{D}^*} [\text{MSE}], \quad \text{MSE} = \| s_{j+1}^* - (\hat{f}(s_j^*, a_j^*) + s_j^*) \|. \tag{5}$$

There are multiple ways to enforce Lipschitz continuity on the learned dynamics function, with varying levels of strength that trade off theoretical guarantees and learning ability. We discusses using (1) global spectral normalization, (2) penalty or (3) slack variables to enforce local Lipschitz continuity.

**Global Lipschitz Continuity via Spectral Normalization**. Follow [39], using spectral norm with coefficient $L$ provides the strongest guarantee that the dynamic model is globally $L$-Lipschitz. Concretely, spectral normalizat [38] normalizes the weights of the neural network following each gradient update.

$$\arg \min_{\hat{f}} E_{s_j^*, a_j^*, s_{j+1}^* \sim \mathcal{D}^*} \left[ \text{MSE} \quad \text{while } W \to W/\max(\frac{||W||_2}{\lambda}, 1) \right] \tag{6}$$

However, spectral normalization enforces *global* Lipschitz bound. It may hinder the model's ability to learn the true dynamics.

**Local Lipschitz Continuity via Sampling-based Penalty**. Following [44], a simple way to relax the global Lipschitz continuity constraint is by penalizing any violation of local Lipschitz constraint

$$\arg \min_{\hat{f}} E_{s_j^*, a_j^*, s_{j+1}^* \sim \mathcal{D}^*} \left[ \text{MSE} + \lambda \cdot \mathbb{E}_{\Delta_s \sim \mathcal{N}} \max \left( \hat{f}'(s_j^* + \Delta_s, a_j^*) - L, 0 \right) \right]. \tag{7}$$

Doing so ensures that the approximate model is mostly $L$ Lipschitz-bounded while being predictive of the transitions in the expert data. This approximate dynamics model can then be used to generate corrective labels. The sampling procedure, $\mathbb{E}_{\Delta_s \sim \mathcal{N}} \max \left( \hat{f}'(s_j^* + \Delta_s, a_j^*) - L, 0 \right)$, is indicative of whether the local continuity constraint is violated for a given state-action pair.

The sampling-based penalty perturbs the data points by some sampled noise and enforces the Lipschitz constraint between the perturbed data and the original using a penalty term in the loss function.

We here propose another approach for local continuity:

**Local Lipschitz Continuity via Slack-variable**. We enforce as much local Lipschitz continuity as possible and, for parts of the space that are discontinuous (e.g., have a very large Lipschitz constant), we do not enforce continuity and would not trust the model outside the data support. We can explicitly allow for a small amount of discontinuity in the learned dynamics model in the same way that slack variables are modeled in optimization problems, e.g., SVM [15] or max-margin planning with slack variables [45]. We can reformulate the dynamics model's learning objective from learning models that maximize likelihood while minimizing the Lipschitz constant.

$$\arg \min_{\hat{f}} \mathbb{E}_{s_j^*, a_j^*, s_{j+1}^* \sim \mathcal{D}^*} \text{MSE} + \lambda_j \cdot Lipschitz(s_j^*, a_j^*) + ||\lambda_j - \bar{\lambda}||_0$$
$$\text{where } ||\lambda_j - \bar{\lambda}||_0 \approx 1 - \exp\left(-\beta|\lambda_j - \bar{\lambda}|\right) \tag{8}$$
$$\text{and } Lipschitz(s_j^*, a_j^*) = \mathbb{E}_{\Delta_s \sim \mathcal{N}} \max \left( \hat{f}'(s_j^* + \Delta_s, a_j^*) - L, 0 \right).$$

To account for small amounts of discontinuity, we introduce a state dependent variable $\lambda_j$ to allow violation of the continuity constraint. We minimize the number of non-zero entries in the slack variables (as noted by the L0 norm, $||\lambda_j - \bar{\lambda}||_0$) to ensure the model is otherwise as smooth as possible. Our practical approximation for the L0 norm method is inspired by [46] and is similar to how robust SVMs and max-margin classifiers deal with outliers.

Intuitively, we expect that spectral normalization tends to work better for simpler environments where the ground truth dynamics are global Lipschitz with some reasonable $L$, whereas the soft constraint should be better suited for data regimes with more complicated dynamics and discontinuity.

### D.2 Generating corrective labels

In Sec. 2.3 we discuss two techniques to generate corrective labels. Depending on the structure of the application domain, one can choose to generate labels either by backtrack or by disturbed actions. Both techniques require solving root-finding equations (Eq. 1 and Eq. 2). To solve them, we can transform the objective to an optimization problem and apply gradient descent.

Eq. 1 specifies the root finding problem for backtrack labels.

$$s_t^* - \hat{f}(s_{t-1}^{\mathcal{G}}, a_t^*) - s_{t-1}^{\mathcal{G}} = 0.$$

Given $s_t^*, a_t^*$ and the learned dynamics function $\hat{f}$, we need to solve for $s_t^{\mathcal{G}}$ that satisfies the equation. We can instead optimize for

$$\arg\min_{s_{t-1}^{\mathcal{G}}} ||s_t^* - \hat{f}(s_{t-1}^{\mathcal{G}}, a_t^*) - s_{t-1}^{\mathcal{G}}|| \tag{9}$$

Similarly, we can transform Eq. 2 to become

$$\arg\min_{s_t^{\mathcal{G}}} ||s_t^{\mathcal{G}} + \hat{f}(s_t^{\mathcal{G}}, a_t^* + \Delta) - s_{t+1}^*|| \tag{10}$$

With access to the trained model $\hat{f}$ and its gradient $\frac{\partial \hat{f}}{\partial s_t^{\mathcal{G}}}$, one can use any optimizer. For simplicity, we use the Backward Euler solver that apply an iterative update $s_t^{\mathcal{G}} \leftarrow s_t^{\mathcal{G}} - s \cdot \frac{\partial \hat{f}}{\partial s_t^{\mathcal{G}}}$ where $s$ is a step size. We repeat the update until the objective is within a threshold $||s_t^{\mathcal{G}} + \hat{f}(s_t^{\mathcal{G}}, a_t^* + \Delta) - s_{t+1}^*|| \leq \epsilon_{opt}$.

### D.3 Using the generated labels

There are multiple ways to use the generated corrective labels. We can augment the dataset with the generated labels and treat them as if they are expert demonstrations. For example, for all experiments conducted in this paper, we train behavior cloning agent using the augmented dataset. Optionally, one can favor the original expert demonstrations by assigning higher weights to their training loss. We omit this step for simplicity in this paper.

Alternatively, one can query a trained imitation learning policy with the generated labels and measure the difference between our generated action and the policy output. This difference can be used as an alternative metrics to evaluate the robustness of imitation learning agents when encountered a subset of out-of-distribution states. However, for a query state, our proposal does not necessarily recover *all* possible corrective actions. We defer exploring alternatives way of leveraging the generated labels to future work.

### D.4 Solving root-finding equation in Generating Labels

To generate corrective labels, both of our techniques need to solve root-finding equations (Eq. 1 and 2). We turn this to an optimization problem: $\arg\min_{s^{\mathcal{G}}} ||s^{\mathcal{G}} + \hat{f}(s^{\mathcal{G}}, a^{\mathcal{G}}) - s^*||$ given $a^{\mathcal{G}}, s^*$. Given $\hat{f}$, there are many ways to solve this optimization problem (e.g., gradient descent or Newton's method). For simplicity, we choose a fast-to-compute and conceptually simple solver, Backward Euler, widely adopted in modern simulators; this lets us use the gradient of the next state to recover the earlier state without iteration. To generate data given $a^{\mathcal{G}}, s^*$, Backward Euler solves a surrogate equation iteratively: $s^{\mathcal{G}} \leftarrow s^* - \hat{f}(s^*, a^{\mathcal{G}})$.

### D.5 Rejection Sample

We can reject the generated labels if the root solver returns an answer that would result in a large error bound, i.e., $||s_t^* - s_t^{\mathcal{G}}|| > \epsilon$, where $\epsilon$ is a hyper-parameter. Rejecting labels that are outside a chosen region lets us directly control the size of the resulting error bound.

## E Experimental Details

We here elaborate on our experiments results over 4 disctinct robotic domains and 14 tasks.

### E.1 The Classic Control: Pendulum and Our Discontinuous Pendulum

We consider the classic control task, the pendulum, where we have access to the ground truth dynamics. We also create a variant, The Discontinuous Pendulum, by inserting a wall that would revert the velocity to bounce the ball back, as shown in Fig. 3a.

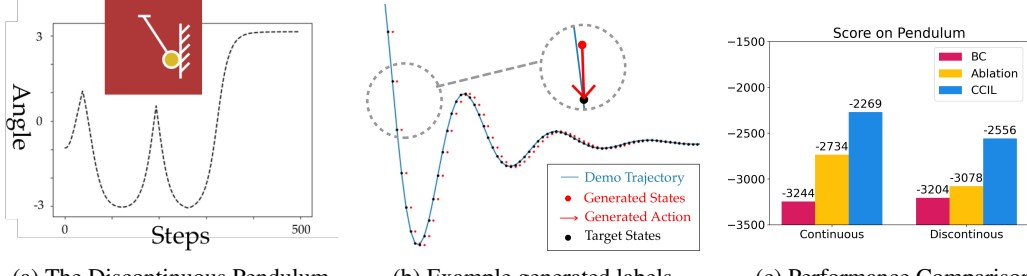

(a) The Discontinuous Pendulum     (b) Example generated labels     (c) Performance Comparison

Figure 3: Evaluation on the Pendulum and Discontinuous Pendulum Task.

**Verifying the quality of the generated labels (Q1)**. We visualize a subset of generated labels in Fig. 3b. Note the generated states (red) are slightly outside the expert support (blue) and that the generated actions are torque control signals, which could be challenging for invariance-based data augmentation methods to generate. To quantify the quality of the generated labels, we use the ground truth dynamics and measure how close our labels can bring the agent to the expert. We observed an average L2 norm distance of $0.02367$ which validates our derived theoretical bound of $0.065$: $K1|\delta| + (1 + K2)|\epsilon| = 12 * 0.0001 + 13 * 0.005$ (Equation 3).

**The Impact of Local Lipschitz Continuity Assumption (Q2)** To highlight how discontinuity in the dynamics affects CCIL, we compare CCIL performance in the continuous and discontinuous pendulum in Fig. 3c: CCIL improved behavior cloning performance even when discontinuity is present, albeit with a smaller boost for discontinuous Pendulum. For ablation, we also tried generating labels using a naive dynamics model (without explicitly assuming Lipschitz continuity) which performed slightly better than vanilla behavior cloning but worse than CCIL, highlighting the importance of enforcing local Lipschitz continuity in training dynamics function for our proposal.

**CCIL improved the performance of imitation learning agent (Q3)**. For both the Pendulum and the discontinuous Pendulum, CCIL outperformed behavior cloning, shown in Fig. 3c.

### E.2    Drone Navigation: High-Frequency Control Task and Sensitive to Noises

Drone navigation is a high-frequency control task and can be very sensitive to noise, making it an appropriate testbed for robustness [39]. We consider an open-source quadcopter simulator, gym-pybullet-drone [47] and design three tasks: hover, circle, fly-through-gate, as shown in Fig. 4.

**CCIL improved performance for imitation learning agent and robustness to noises (Q2)**. On all three tasks, CCIL outperformed behavior cloning by a large margin, achieving near-expert performance across 3 random seeds. We injected observation and action noises to further evaluate the robustness of the learner agent and observed that CCIL achieved consistent performance despite the injected disturbance whereas the default behavior cloning method suffered from high variance of performance.

### E.3    Driving with Lidar: High-dimensional state input

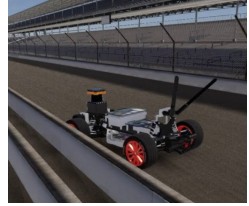     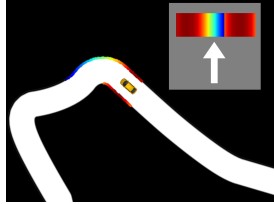

| Method | Succ. Rate | Avg. Score |
|---|---|---|
| Expert | 100.0% | 1.00 |
| BC | 31.9% | $0.58 \pm 0.25$ |
| NoiseBC | 39.2% | $0.63 \pm 0.27$ |
| CCIL | **56.4%** | **0.75** $\pm 0.25$ |

Figure 5: F1tenth     Figure 6: LiDar POV     Table 2: Performance on Racing

We apply CCIL to high-dimensional input states with complex discontinuities and conduct experiments in the F1tenth simulator employing a LiDAR sensor as input (Fig. 5). We design a racing track such that, at the beginning of each trajectory, the car is placed at a random location on the track. It

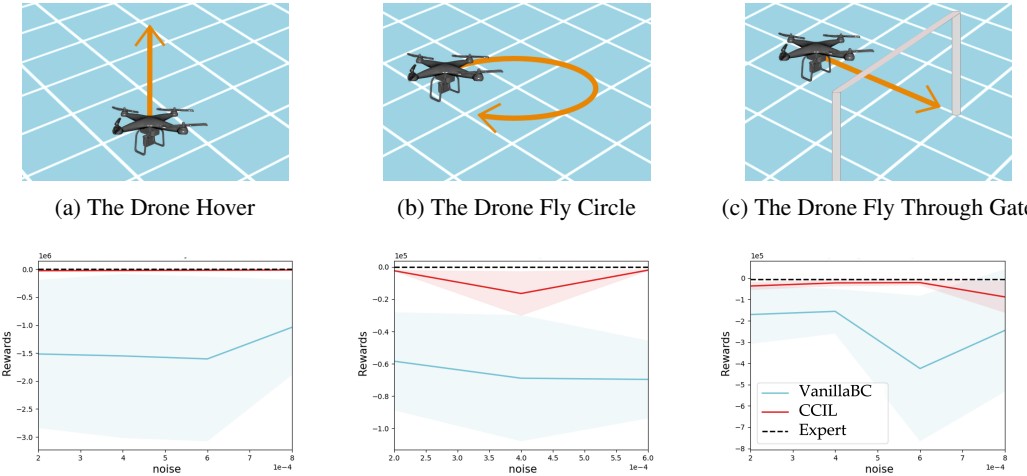

Figure 4: The Drone Navigation Tasks and Evaluation Results

Table 3: Evaluation results for Mujoco and Metaworld tasks with noise disturbances. In the face of varying discontinuity from contacts, CCIL can still outperform vanilla behavior cloning or at least achieve comparable performance. It is the leading agent on 4 out of 8 tasks. NoiseBC is the winner on 1 task (Ant). The rest 3 tasks (ButtonPress, CoffeePush, DrawerClose) observe ties between CCIL and NoiseBC.

| | **Mujoco** | | | | **Metaworld** | | | |
| | Hopper | Walker | Ant | Halfcheetah | CoffeePull | ButtonPress | CoffeePush | DrawerClose |
|---|---|---|---|---|---|---|---|---|
| Expert | 3234.30 | 4592.30 | 3879.70 | 12135.00 | 4409.95 | 3895.82 | 4488.29 | 4329.34 |
| VanillaBC | 2902.78 ± 689.64 | 3810.63 ± 828.23 | 1646.24 ± 202.71 | 3872.82 ± 460.09 | 3552.59 ±233.41 | 3693.02 ± 104.99 | 1288.19± 746.37 | 2809.56±439.70 |
| NoiseBC | 1563.56 ± 1012.02 | 2893.21 ± 1076.89 | **3160.51**± 48.68 | 2044.24 ±291.59 | 3072.86 ± 785.91 | 3663.44±63.10 | 2551.11± 857.79 | 4226.71± 18.90 |
| CCIL | **3102.25** ± 309.25 | **4605.26** ± 129.02 | 2073.60 ± 217.97 | **4182.15** ± 501.44 | **4168.46** ± 192.98 | 3775.22±91.24 | 2484.19± 976.03 | 4145.45± 76.23 |

needs to use the LiDAR input to decide on speed and steering, earning scores for driving faster or failing by crashing. We evaluate each agent over 100 trajectories multiplied by 10 random seeds.

**CCIL could improve the performance of imitation learning agent for high-dimensional state inputs (Q2).** Table. 2 shows that CCIL demonstrated an empirical advantage over all other agents, achieving fewer crashes and higher scores.

### E.4 Locomotion and Manipulation: Diverse Scenes with Varying Discontinuity

Manipulation and locomotion tasks commonly involve complex forms of contacts, raising considerable challenges for learning dynamics models and for our proposal. We evaluate the applicability of CCIL in such domains: we consider 4 tasks from the MuJoCo locomotion suites: Hopper, Walker2D, Ant, HalfCheetah and 4 tasks from the MetaWorld manipulation suites: CoffeePull, ButtonPress, CoffeePush, DrawerClose. During evaluation, we add a small amount of randomly sampled Gaussian noise to the sensor (observation state) and the actuator (action) to simulate the real-world conditions of robotics controllers and to test the robustness of the agents.

**CCIL outperforms vanilla behavior cloning or at least achieves comparable performance even when varying form of discontinuity is present (Q3).** On 4 out of 8 tasks in MuJoCo and MetaWorld, CCIL outperforms all other baselines. Across all tasks, CCIL at least achieves comparable results to vanilla behavior cloning, shown in Table. 3.

### E.5 Summary

Through our extensive evaluations, we answered **Q1** by validating the theoretical bounds we derived on the classic control problem of Pendulum; **Q2** by evaluating over classic control task, drone navigation and high-dimensional driving tasks. We observed that CCIL consistently boosted the performance and robustness of behavior cloning agents; **Q3** by running ablation studies on the Pendulum and including experiments in domains that have complicated dynamics. We found that discontinuity in the dynamics would add challenge to training dynamics function and to our proposal.

However, CCIL could still improve behavior cloning agent with a appropriately-trained dynamics model or, in the worst case, achieves comparable performance to vanilla behavior cloning.

### E.6 Reproducing our experiments

We provide details to reproduce our experiments, including environment specification, expert data, parameter tuning for our proposal and details about the baselines. We will also open source the code and the configuration we use for each experiment, once the proposal is published.

### E.7 Environment and Task Design

We conduct experiments on 4 different domains and 8 robots, including the pendulum, a drone, a car, four robots for locomotion and one robot arm for manipulation. We consider 18 tasks: the pendulum, a modified pendulum swing task with discontinuity, three drone navigation tasks (fly-through, circle, hover), one LiDar racing task on F1tenth, four MuJoCo tasks (Hopper, HalfCheetah, Ant, Walker2D) and 8 MetaWorld tasks (coffee-pull, coffee-push, button-press-topdown, drawer-close, drawer-open, window-close, push, soccer). The drone, F1tenth, MuJoCo and Metaworld environments are from open source implementations. We will describe how we set up the Pendulum environment and how we modify it for testing our method with discontinuity.

**Pendulum Formulation.** The pendulum environment asks a policy to swing a pendulum up to the vertical position by applying torque. The properties of the system are controlled by the constants $g$, the gravitational acceleration, and $l$, the length of the pendulum. In all experiments we take $g = 9.81$ and $l = 1$.

A pendulum's state is characterized by $\theta$, the current angle, and $\dot{\theta}$, the current angular velocity. To avoid any issues regarding angle representation, we do not directly store $\theta$ in the state representation; instead, we parameterize the state as $s = \begin{bmatrix} \sin \theta & \cos \theta & \dot{\theta} \end{bmatrix}^T$. A policy can control the system by applying torque to the pendulum, which we represent as a scalar $a$, which is clamped to the range $[-3, 3]$.

The continuous time dynamics function is given by:

$$\frac{ds}{dt} = f(s, a) = \begin{bmatrix} \dot{\theta} \cos \theta \\ -\dot{\theta} \sin \theta \\ -\frac{g}{l} \sin \theta + a \end{bmatrix}.$$

This continuous dynamics model is then discretized to a timestep of 0.02 seconds using RK4. Additionally, although not required by the algorithms we study, we create the following reward function. where $\theta$ is the normalized pendulum angle in the range $[0, 2\pi)$ to metricize policy performance:

$$r(s, a) = -\frac{1}{2} \left\| \begin{bmatrix} \theta - \pi \\ \dot{\theta} \end{bmatrix} \right\|_2^2 - \frac{1}{2} a^2.$$

**Expert Formulation for Pendulum.** We formulate the expert policy using a combination of LQR and energy shaping control, where LQR is applied when the pendulum is near the top and energy-shaping is applied everywhere else. Note that the LQR gains were calculated by linearizing the dynamics around $\theta = \pi$, along with the cost function $c(s, a) = -r(s, a)$. So, the expert policy has the form:

$$\pi_e(s) = \begin{cases} -20.11(\theta - \pi) - 7.08\dot{\theta} & \text{if } |\theta - \pi| < 0.1 \\ -\dot{\theta} \left( \frac{1}{2}\dot{\theta}^2 - 9.81 \cos \theta - 9.81 \right) & \text{otherwise.} \end{cases}$$

**Discontinuous Pendulum** . We create a fixed wall in the Pendulum environment to create local discontinuity. When the ball hits the wall, we *revert* the sign of its velocity, creating a discontinuity in the dynamics.

| Environment | Trajectories |
|---|---|
| Pendulum | 50 |
| Discontinuous Pendulum | 500 |
| F1tenth Racing | 1 |
| Drone, all three tasks | 5000 |
| MuJoCo - Ant | 10 |
| MuJoCo - Walker2D | 20 |
| MuJoCo - Hopper | 25 |
| MuJoCo - HalfCheetah | 50 |
| MetaWorld, all tasks | 50 |

Table 4: Number of expert demonstration trajectories used in our experiments. We limit the amount of expert data to avoid making the task trivially solvable by naive behavior cloning.

### E.8 Demonstration Data

To feed expert data to train imitation learning agents, we design expert policies for the pendulum. For all other environments, we use the expert data from the D4RL dataset [48]. For drone environments, we first generate a bunch of via points alongside the target trajectories and then use a low-level PID controller to hit the via points one by one.

We note that it is possible to solve most tasks with naive behavior cloning if we feed them with a sufficient number of demonstrations. We thus limit the number of demonstrations we use for all tasks, as shown in Table. 4.

### E.9 Parameter Tuning.

Our proposal first trains a dynamics model and has hyperparameters: $\bar{L}$ (desired local Lipschitz smoothness per NN layer), $\lambda$ (weight of the Lipschitz penalty) and $\sigma$ (size of perturbation for estimating local Lipschitz continuity). We first fit a dynamics model without enforcing any Lipschitz smoothness, to obtain an average prediction error for reference. To enforce local Lipschitz continuity, we then adopt the sampling-based penalty and train a series of dynamics models by sweeping parameters, $\bar{L} = [2, 3, 5, 10]$, soft dynamics $\lambda = [0.3, 0.5]$ and $\sigma = [0.0001, 0.0003, 0.0005]$. In environments with many discontinuities (MuJoCo and MetaWorld tasks), we use the slack variable to train the dynamics model. Following Eq. 8, we introduce an additional learnable variable, the slack variable. We choose a relatively smaller $\beta$ (in our case we choose 0.1), to avoid the explosion on gradients of zero norm estimation. The slack variable is a state-conditioned network that could be easily modeled as a two-layer MLPs(64,32). As to the baseline number $\bar{\lambda}$, we choose it to be 0.1.

To choose a dynamics model for generating labels, we will follow Theorem. C.1 and Theorem. C.2. We note that the local Lipschitz bound of a neural network is the product of the Lipschitz bound of each layer. Given that we are using two-layer NN to train our dynamics model, $L = \bar{L}^2$. We denote the empirical prediction error on the validation set for each trained dynamics model as $\epsilon$. We choose the best dynamics model that has the smallest error bound. For Theorem. C.1, we optimize for $\epsilon + 2L \cdot ||C||$ where $C$ is a constant that we pick to be the average $s_{t+1}^* - s_t^*$ across the training data. For Theorem. C.2, we optimize for $0.001 \cdot L + (1 + L)\epsilon$.

To generate corrective labels, we pick the dynamics model, the size of the perturbation (Sigma) and the rejection threshold (Epsilon). Empirically, Sigma = 0.00001 and Epsilon = 0.01 worked for all our environments. It is also possible to fine-tune Epsilon, the rejection threshold, for each task and each trained dynamics model to optimize the error bound. In our experiments, we omit this step for simplicity.

After generating corrective labels, we use two-layer MLPs (64,64) plus ReLu activation to train a Behavior Cloning agent with both original and augmented data.

### E.10  Baselines

We evaluate the following algorithms to gain a thorough understanding of how our proposal compares to relevant baselines.

- EXPERT: For reference we plot the theoretical upper bound of our performance, which is the score achieved by an expert during data collection.
- BC: a naive behavior cloning agent that minimizes the KL divergence on a given dataset.
- NOISEBC: a modification to naive behavior cloning that injects a small disturbance noise to the input state and reuses the action label, as described in [9].
- CCIL: our proposal to generate high-quality corrective labels.

