# OpenReview forum: "CCIL: Continuity-Based Data Augmentation for Corrective Imitation Learning"
_robot-learning.org/CoRL/2023/Workshop/OOD — OOD Workshop @ CoRL 2023_

### Official Review · Reviewer_kngv · 2023-10-16
**Accept**

**Rating:** 6
**Confidence:** 4

**Review:**

In this paper, the authors present a technique to generate "corrective labels" to augment a dataset for imitation learning. They use expert demonstrations to learn a dynamics model and then enforce local Lipschitz continuity in the model using slack variables. This local continuity allows them to generate perturbed <state, action, next state> tuples that are in the neighborhood of (but outside) the demonstration dataset. These labels redirect agents from unfamiliar states back to familiar ones. They show that the resulting algorithm, CCIL, outperforms vanilla behavior cloning in simulation tasks such as drone navigation, car racing using lidar, Mujoco locomotion, and Metaworld manipulation.

The paper is well-written, the approach is technically sound, and the results are impressive. The paper does not directly address the problem of incorporating OOD inputs to the pipeline, but can easily be extended so that the generated labels are selected to perturb an OOD input encountered during deployment towards a familiar state. The one downside is that the method assumes at least local Lipschitz continuity, which may be violated when an input is from a wholly new distribution. Nonetheless, topics such as dataset augmentation and reasoning about corrective labels are important to the discussion about OOD. I recommend this paper be accepted.

---

### Official Review · Reviewer_9Gko · 2023-10-16
**Some difficulties on the assumptions of Lipschitz continuity but an interesting idea**

**Rating:** 7
**Confidence:** 4

**Review:**

This paper follows the promising philosophy that robustness to OOD inputs involves leveraging prior knowledge or assumptions about the problem. This allows for the behavior cloning data augmentation to be done without the expensive step of querying an expert again. One can imagine how this work can be extended to involve other assumptions that one may have on a dynamics model; for example, there may be an underlying differential equation involved. Also, could this method be used online somehow when the agent is faced with distribution shift?

Some questions are: how would one choose a Lipschitz constant / state-dependent Lagrange multiplier? How would the data augmentation work near discontinuous states in the dynamics? Also, from the way the model is trained there do not seem to be any guarantees that the model will indeed abide by local Lipschitz continuity.

For real systems, the applicability of this method seems to be dependent on the sampling rate. For example, if a robot is moving too fast, it can leave the OOD “buffer” region before it gets a chance to apply the corrective action.

Also, why does the dataset not get augmented for all states and actions near the expert dataset? Instead of only corrective actions, there can be “trajectories” in the buffer zone allowing the agent to come back to the support of the expert dataset.

Overall, this work has an interesting approach to the problem of coming “back in-distribution” when OOD by augmenting the dataset with corrective actions. It will lead to good discussions that we should have as a community on underlying assumptions on dynamics (ex. continuous, hybrid, etc.) and how to incorporate the relevant tools from control theory.

---

### Decision · Program_Chairs · 2023-10-17

**Decision:**

Accept

**Comment:**

We agree with the reviewers’ assessment that this work is technically sound and will contribute to productive, topical discussions at the 2023 Workshop on OOD Generalization in Robotics. In particular, we appreciate that this work highlights how one can leverage domain knowledge to improve generalization outside the support of the training data, using data-augmentations as the core technique.
We recommend the authors incorporate the reviewers’ feedback into their camera-ready submission to further improve their manuscript.